# The Use of Automated Atrial CMR Measures and a Novel Atrioventricular Coupling Index for Predicting Risk in Repaired Tetralogy of Fallot

**DOI:** 10.3390/children10020400

**Published:** 2023-02-18

**Authors:** Megan Gunsaulus, Alejandra Bueno, Carley Bright, Katelyn Snyder, Nikkan Das, Craig Dobson, Mark DeBrunner, Adam Christopher, Arvind Hoskoppal, Christopher Follansbee, Gaurav Arora, Laura Olivieri, Tarek Alsaied

**Affiliations:** Pediatric Cardiology UPMC Children’s Hospital of Pittsburgh, 4401 Penn Avenue, 5th Floor Faculty Pavilion, Pittsburgh, PA 15224, USA

**Keywords:** tetralogy of Fallot, cardiac MRI, diastolic function, atrial size, right atrioventricular coupling index (RACI)

## Abstract

Atrial size and function have been recognized as markers of diastolic function, and diastolic dysfunction has been identified as a predictor of adverse outcomes in repaired tetralogy of Fallot (rTOF). This was a retrospective single-center study with the objective of investigating the use of atrial measurements obtained via CMR for predicting outcomes in rTOF patients. Automated contours of the left and right atria (LA and RA) were performed. A novel parameter, termed the Right Atrioventricular Coupling Index (RACI), was defined as the ratio of RA end-diastolic volume to right ventricle (RV) end-diastolic volume. Patients were risk-stratified using a previously validated Importance Factor Score for the prediction of life-threatening arrhythmias in rTOF. Patients with a high-risk Importance Factor Score (>2) had a significantly larger minimum RA volume (*p =* 0.04) and RACI (*p =* 0.03) compared to those with scores ≤2. ROC analysis demonstrated RACI to be the best overall predictor of a high-risk Importance Factor Score (AUC 0.73, *p =* 0.03). Older age at the time of repair and a diagnosis of pulmonary atresia were associated with a larger RACI. Automated atrial CMR measurements are easily obtained from standard CMRs and have the potential to serve as noninvasive predictors of adverse outcomes in rTOF.

## 1. Introduction

Over the past several decades, there has been significant improvement in surgical techniques and post-operative care for patients with tetralogy of Fallot (TOF). This has led to a growing population of patients living with repaired TOF (rTOF). Despite improvements in early mortality, there is still a relatively high incidence of morbidity and mortality late after repair, beginning primarily in the third decade of life [1,2]. Late complications include ventricular dysfunction, atrial and ventricular tachycardia, and sudden death [3]. These problems have historically been attributed to progressive right ventricle (RV) dysfunction resulting from chronic pressure and/or volume load [4].

RV size and systolic function have therefore been the primary areas of focus when assessing the timing of pulmonary valve replacement (PVR) and long-term outcomes [5]. However, previous studies have demonstrated that increased RV volume is not consistently associated with sustained ventricular tachycardia or increased mortality [6]. This highlights the need for additional noninvasive measures to serve as predictors of adverse clinical outcomes and inform clinical decision making. See Figure 1 for visual abstract.

Diastolic dysfunction has emerged as an important predictor of adverse outcomes in multiple forms of congenital heart disease [7]. In rTOF, increased left ventricular end-diastolic pressure measured by cardiac catheterization is a known risk factor for ventricular tachycardia and sudden cardiac death [8]. Another important risk factor for sudden death in rTOF is atrial tachycardia, which may be secondary to atrial dilation [6]. Both atrial size and atrial function have been recognized as surrogate parameters for assessing diastolic dysfunction [9,10]. CMR provides important insights into the risk of adverse clinical outcomes in rTOF, and multiple risk scores have been developed based on a combination of clinical and CMR parameters. One such risk score is the “Importance Factor Score”, created by Atallah J et al. A high-risk Importance Factor Score was found to be associated with life-threatening arrhythmic events with a sensitivity of 88% and a specificity of 68% [11].

Another recently devised CMR prognostic tool, the left atrioventricular coupling index (LACI), has been evaluated as part of the Multi-Ethnic Study of Atherosclerosis (MESA) [12]. LACI is defined as the ratio between LA end-diastolic volume and LV end-diastolic volume. It was created to determine whether the close physiological relationship between the LA and LV could serve as a primary prevention tool in the early detection of cardiovascular disease. LACI has been shown to serve as a strong predictor for the incidence of heart failure, atrial fibrillation, cardiovascular disease, and coronary heart disease death in healthy adult populations [12,13,14]. A novel right atrioventricular coupling index can potentially help with risk stratification in patients with rTOF.

There has been limited investigation intof the use of atrial measurements obtained via CMR for predicting outcomes in rTOF patients. Thus, we examined the relationship between left and right atrial volumes and function and ventricular parameters from CMR. We also determined the association of atrial measurements and atrioventricular coupling with the risk of adverse long-term outcomes using an established risk score. Additionally, we assessed the association of atrial CMR parameters with pulmonary valve replacement.

## 2. Materials and Methods

### 2.1. Patients

A database search identified all rTOF individuals older than 10 years of age who had a CMR study at the Children’s Hospital of Pittsburgh or the UPMC Presbyterian Hospital between 2011 and 2020. A total of 63 patients were included in the study. This study was approved by the University of Pittsburgh Institutional Review Board and was conducted in compliance with the Health Insurance Portability and Accountability Act. The requirement for informed consent was waived due to the retrospective nature of the study.

### 2.2. Clinical Data

Demographic and clinical data were obtained from electronic medical records. These included patients’ surgical and procedural histories as well as long-term outcomes, such as death, hospital admissions, and the need for implantable cardioverter defibrillator (ICD) placement.

### 2.3. CMR

CMR studies were performed using GE HDxT 1.5 Tesla scanners (Medical Systems, Milwaukee, Wisconsin) or Siemens Sola 1.5 Tesla scanners (AG, Munich, Germany). Briefly, ventricular assessment was performed using electrocardiographically gated, balanced steady-state free precession (bSSFP) cine CMR in vertical and horizontal ventricular long-axis planes. Standard 2- and 4-chamber cine images were imported into post-processing software (CVi42, Circle Cardiovascular Imaging Inc., Calgary, Canada). Ventricular volumes and function were measured by tracing the endocardial and epicardial borders on each short-axis bSSFP cine slice at end-diastole (maximal volume) and end-systole (minimal volume). The pulmonary regurgitation fraction was computed based on standard 2-dimensional phase-contrast imaging positioned in the main pulmonary artery. Automated biplane atrial function evaluation was performed to measure left atrial (LA) volumes; contours were adjusted manually as needed. The right atrial (RA) volumes were measured from monoplane 4-chamber views (Figure 2). LA and RA maximal (end-systolic) and minimal (end-diastolic) volumes were calculated according to the biplane area–length and the single plane area–length methods, respectively. LA and RA atrial ejection fractions were also computed. If a patient had multiple CMR studies, the most recent study was used for analysis. All CMR measurements were performed by a single pediatric cardiologist. To account for interobserver variability, an additional pediatric cardiologist repeated the measurements for several patients chosen at random and made comparable findings.

### 2.4. Risk Scores

A previously established Importance Factor Score was calculated for each patient. This score includes the following variables: the presence of symptoms, LV dysfunction, RV dysfunction, RV pressure load, age at repair, history of shunt pre-repair, complexity of repair, QRS duration ≥180 ms, and evidence of ventricular tachycardia. Left ventricular dysfunction (moderate to severe) was considered as an LVEF less than 45%, as previously defined. Right ventricular dysfunction (moderate to severe) was defined as an RVEF less than 40%. Moderate-to-severe RV volume load was defined as an RVEDVi greater than 130 mL/m^2^. The TOF repair was considered complex if it required an RV-to-pulmonary-artery (PA) conduit rather than a transannular patch. This score was modified slightly based on the available data. We did not include the presence of moderate-to-severe RV pressure load, as these data were not available; this parameter was worth a maximum of 1 point and minimally contributed to the total score. An Importance Factor Score >2 was considered high risk. 

A new risk score was also calculated for each patient and was termed the “RV-Independent Score”. This score was similar to the Importance Factor Score described above, with 2 exceptions. First, right ventricular end-diastolic volume was not included in the formula in order to investigate whether RA volume correlated with the risk score independently of RV volume. Second, a QRS duration of ≥120 ms was used as a cutoff point instead of 180 ms because our population was relatively young and none of our patients had a QRS duration >180 ms. The cutoff point of 120 ms was chosen given that this is the QRS duration for diagnosis of a right bundle branch block. Additionally, approximately one-third of our population had a QRS duration below 120 ms. An RV-Independent Score >2 was considered high risk.

### 2.5. Right Atrioventricular Coupling Index

Based on prior studies demonstrating the prognostic value of LACI [12,13,14], we sought to evaluate the relationship between RA and RV in rTOF patients by creating a right atrioventricular coupling index (RACI). RACI was defined for each subject as the ratio between the RA end-diastolic volume and the RV end-diastolic volume assessed by CMR.

### 2.6. Statistical Analysis

Continuous data were summarized as medians (25th quartile, 75th quartile), and categorical data were summarized as N (%). The Kruskal–Wallis and Mann–Whitney *U* tests were used for comparison of non-normally distributed independent categorical variables and dependent continuous variables. The *t*-test was used for normally distributed variables. Univariable associations between continuous variables were determined using the Spearman correlation. Receiver-operating characteristic (ROC) curves were generated to evaluate the ability of CMR parameters to predict a high-risk Importance Factor Score. Statistical analyses were performed using commercially available software (JMP, version 14 (SAS Institute) and SPSS v27.0 (SPSS, Inc., Chicago, IL)). Two-tailed *p* < 0.05 was considered statistically significant.

## 3. Results

### 3.1. Baseline Characteristics

Sixty-three patients were included in the analysis. The median age at the time of TOF repair was 0.5 (interquartile range (IQR): 0.4, 1.0) years, and the median age at the time of CMR was 16.4 (IQR: 12.9, 18.7) years. CMR data were notable for a median indexed right ventricle end-diastolic volume (RVEDVi) of 119.1 (IQR: 95.5, 138.3) mL/m^2^ and a median pulmonary regurgitation fraction of 39 (IQR: 19, 52) percent. The median RACI was 15.0 (IQR 10.5–22.1) percent (Table 1).

### 3.2. Associations between Atrial CMR Measurements and Patient Characteristics

Older age at the time of TOF repair was associated with a larger maximum RA volume (*p* = 0.04) and RACI (*p* = 0.03) (Table 2). Patients with pulmonary atresia had a significantly larger minimum RA volume and RACI compared to those with a diagnosis of pulmonary stenosis (*p* = 0.03 and 0.02, respectively). The remainder of the atrial CMR measurements did not significantly differ between these diagnoses. There were no significant differences in atrial CMR measurements between patients with and without a history of a BTT shunt (Table 3).

### 3.3. Associations between Atrial CMR Measurements and Ventricular CMR Parameters

The minimum and maximum biatrial volumes were positively associated with a larger LVEDVi and LVESVi (Figure 3). RA ejection fraction was positively associated with a higher LVEF (*p* = 0.03), and LA ejection fraction was positively associated with both a higher LVEF (*p* = 0.02) and RVEF (*p* = 0.02) (Figure 4). A larger RACI was associated with a smaller RA ejection fraction (*p* = 0.0007) and a reduced pulmonary regurgitation fraction (*p* = 0.0002) (Table 2).

### 3.4. Risk Scores

The Importance Factor Scores ranged from a minimum of 0 to a maximum of 7, with a median value of 1. Patients at high risk (Importance Factor > 2) had a significantly larger minimal RA volume and RACI compared to patients with an importance factor ≤2 (19.8 (IQR: 14.3, 28.9) vs. 16.6 (IQR: 11.3, 23.2) mL/m^2^, p = 0.04 and 18.6 (IQR: 11.8, 26.8) vs. 13.5 (IQR: 9.9, 19.6) percent, *p* = 0.03, respectively) (Table 4, Figure 5). ROC analysis of various quantitative CMR measures demonstrated RACI to be the best overall predictor of a high-risk Importance Factor Score (AUC 0.73, *p* = 0.03), superior to RVEF and RVEDVi (Figure 6).

The RV-Independent Score ranged from a minimum of 0 to a maximum of 8 with a median value of 1. Patients with a high-risk RV-Independent Score (>2) had a significantly larger minimal RA volume and RACI compared to patients with a score ≤2 (20.6 (IQR: 14.7, 28.9) vs. 15.4 (IQR: 11.3, 22.7) mL/m^2^, *p* = 0.02 and 19.5 (IQR: 13.4, 26.8) vs. 13.1 (IQR: 9.8, 18.9) percent, *p* = 0.02, respectively) (Table 4, Figure 5).

With regards to long-term outcomes, patients had a median follow-up time of 5.8 years from the time of CMR. There were no deaths in this cohort. Only one patient required ICD placement and hospitalization for arrhythmias; this patient had a maximum LA and RA volume greater than the 70th percentile and an RACI greater than the 98th percentile. One patient required hospitalization for management of congestive heart failure symptoms; this patient had a minimum and maximum RA volume greater than the 90th percentile, with an RA EF less than the 5th percentile and an RACI greater than the 85th percentile.

### 3.5. Effects of PVR on Atrial Parameters

Thirty-one patients (49%) underwent PVR (6 patients before CMR and 25 patients after CMR). For the six patients who underwent CMR following PVR, the average length of time from PVR to CMR was 6.6 years. When comparing the six patients who had undergone PVR prior to the CMR with the rest of the cohort, the maximum RA volumes were larger in the patients with PVR (54.9 (IQR: 29.6, 61.7) vs. 27.6 (IQR: 20.5, 35.9) mL/m^2^, *p* = 0.03). When comparing the 25 patients who underwent PVR after the CMR with those who did not, those who underwent PVR had a significantly lower LA ejection fraction (57.3 (IQR: 46.6, 62.4) vs. 60.4 (IQR: 47.9, 71.6) percent, *p* = 0.03). None of the other parameters were different between the groups.

## 4. Discussion

Our study evaluated the association of automated atrial CMR measurements with patient characteristics, ventricular CMR parameters, and timing of PVR. We also sought to determine the relationship between these atrial parameters and adverse outcomes via an Importance Factor Score surrogate. We found that patients with a high-risk Importance Factor Score (>2) and a high-risk RV-Independent Score (>2) had a larger minimum RA volume and RACI compared to those with scores ≤2. RACI had the highest area under the curve for prediction of a high-risk Importance Factor Score. Older age at the time of TOF repair and a diagnosis of pulmonary atresia were associated with a larger RA volume and a larger RACI. Larger LA and RA volumes were associated with a larger LVEDVi and LVESVi. Finally, patients post-PVR had a larger maximum RA volume compared to those without PVR at the time of CMR. Patients who underwent PVR after CMR had significantly lower LAEF according to CMR compared to subjects who did not ultimately undergo PVR.

Similar to our study, Ait Ali et al. showed that RA dilation was associated with all-cause mortality, aborted sudden cardiac death, and sustained VT in a cohort of 165 patients with rTOF [15]. However, their study used manual contouring of the atria from the short axis. This process requires the acquisition of additional short-axis images, necessitating extra scanning and post-processing time. In contrast, automated atrial CMR measurements are very simple to perform. They require no additional scanning time and minimal post-processing time. CMRs are routinely obtained in this population to assess right heart size and function, so the patient is unlikely to require any additional testing to acquire these measurements. There is limited ability to compare our atrial measurements with normalized values by age due to a paucity of pediatric data. However, the atrial measurements in our cohort appear to be similar to the previously established standard values in healthy adult populations [16].

In this study, we used a previously validated Importance Factor Score as a surrogate for long-term adverse outcomes. We also developed a new risk score modeled on this established score called the RV-Independent Score. This score excluded RVEDVi in order to investigate whether atrial parameters remained significant independently of RV volume. In addition, we adjusted the QRS duration cut off from ≥180 ms to ≥120 ms, which is the QRS duration for a diagnosis of a right bundle branch block. Multiple previous studies, including our recent meta-analysis, have demonstrated that a QRS duration >180 ms and increased RV end-diastolic volume are not associated with mortality or sudden cardiac death in the current era. However, QRS duration as a continuous variable was associated with mortality in rTOF [17]. Previously established risk scores have primarily only targeted older, higher-risk rTOF patients [18]; we aimed to risk-stratify a younger, healthier rTOF population such as the cohort in this study. The performance of the RV-Independent Score requires validation in a larger cohort including patients with adverse outcomes.

We also evaluated the concept of atrioventricular coupling in this study. Our results demonstrate that both RA and LA volumes correlate with LV volume, and RA and LA ejection fractions correlate with LV ejection fraction. Interestingly, we did not find a similar relationship between RA volume and ejection fraction with RV volume and ejection fraction. This is likely due to the presence of pulmonary regurgitation in many of our patients, which is known to affect RV volume and function in rTOF [19,20].

RACI was used as a new predictive tool in our study based on previous findings demonstrating the value of LACI for predicting cardiovascular outcomes in adult populations. RACI relates RA to RV volume, thereby providing a measurement that acknowledges the important interactions that exist between RA and RV performance [21]. In this study, a larger RACI was associated with both of the above risk scores. This is not surprising because a larger RA volume can be seen in patients with RV diastolic dysfunction related to RV fibrosis, which is known to increase the risk of adverse outcomes.

RACI is also inversely related to RA ejection fraction and pulmonary regurgitation (PR) fraction. This latter finding is likely secondary to the known physiological changes associated with chronic PR, which leads to progressive RV enlargement and dysfunction [22]. This in turn results in a smaller RACI because RVEDVi is the denominator of this ratio. Overall, this RACI measurement is easy and fast to perform and has the potential to serve as an additional non-invasive tool for risk-stratifying patients with rTOF. The addition of this atrioventricular coupling index allows for increased attention to diastolic dysfunction and may provide a crucial tool for the risk stratification of rTOF patients.

Although this study demonstrates important findings related to the use of CMR atrial parameters in the rTOF population, there are several inherent limitations. These limitations include its retrospective design and relatively small sample size derived from a single center. Larger, multi-center studies will be needed to validate our results. This would allow for a more direct assessment of the relationship between atrial CMR parameters and long-term outcomes rather than using a surrogate risk score. We used an established Importance Factor Score that has previously been found to be associated with malignant arrhythmias in rTOF patients in order to mitigate this limitation. An additional limitation is the understanding that CMR may not be an accessible test in some settings. It may be useful to evaluate atrial and RACI measurements obtained via echocardiography, particularly in the setting of increasing utilization of three-dimensional echocardiography. Finally, patients with epicardial leads were excluded from this study, as CMR is not deemed safe for this population.

## 5. Conclusions

The results of this study demonstrate that automated atrial volumes and ejection fractions have the potential to serve as additional non-invasive predictors of adverse clinical outcomes in rTOF. This study also highlights the potential benefits of a novel RACI parameter for evaluating atrioventricular interactions and risk-stratifying patients. Patients with a high RACI are at higher risk and may require closer monitoring for the development of adverse outcomes.

## Figures and Tables

**Figure 1 children-10-00400-f001:**
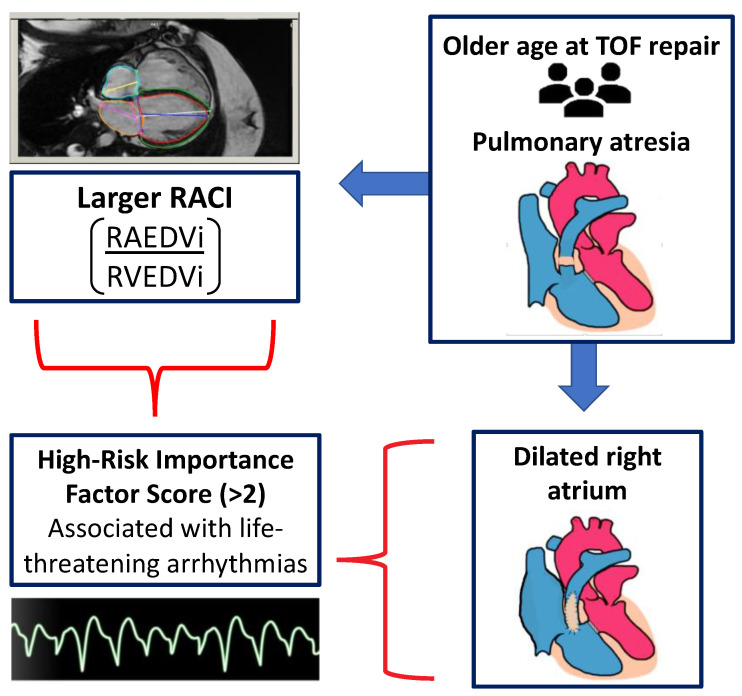
Visual abstract summarizing the primary findings of this paper.

**Figure 2 children-10-00400-f002:**
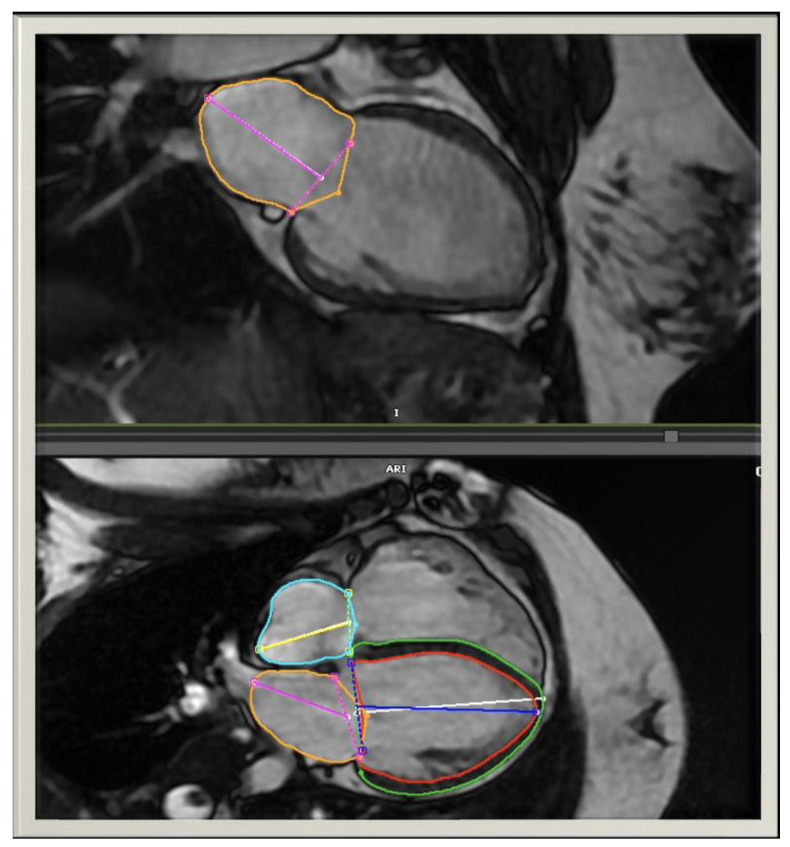
Automated LA volumes were measured in the biplane view and adjusted manually as needed. The RA volumes were measured in monoplane 4-chamber views.

**Figure 3 children-10-00400-f003:**
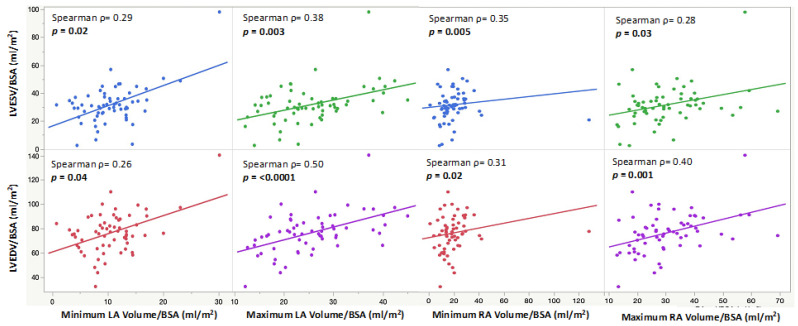
Scatter plots showing indexed minimum and maximum LA and RA volumes and their positive correlations with indexed end-diastolic and end-systolic LV volumes.

**Figure 4 children-10-00400-f004:**
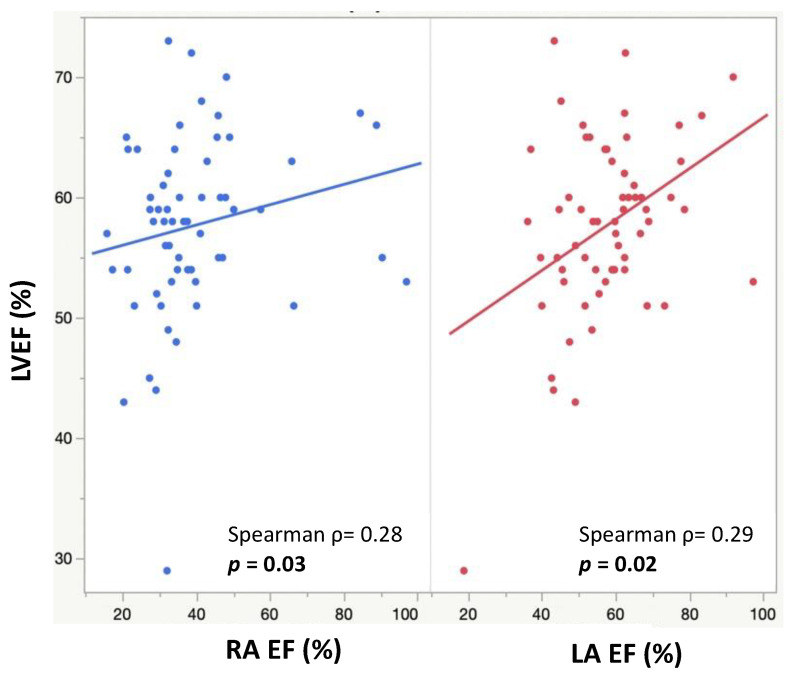
Scatter plots showing RA EFs (blue) and LA EFs (red) and their positive correlations with LVEFs.

**Figure 5 children-10-00400-f005:**
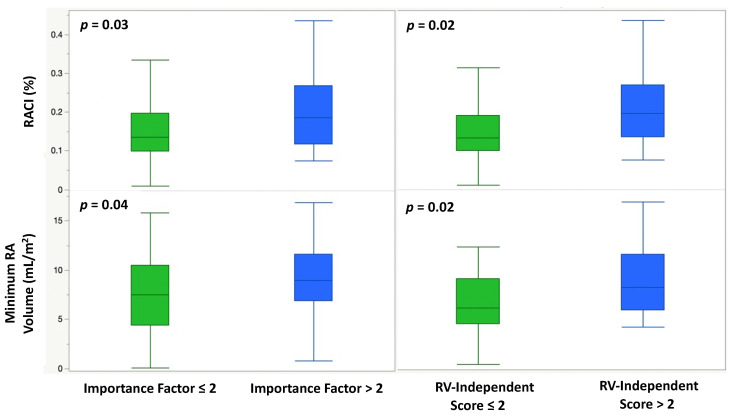
Box-and-whisker plots demonstrating the association of a high-risk Importance Factor Score (>2) and a high-risk RV-Independent Score (>2) with RACI (%) and indexed minimum RA volume (mL/m^2^).

**Figure 6 children-10-00400-f006:**
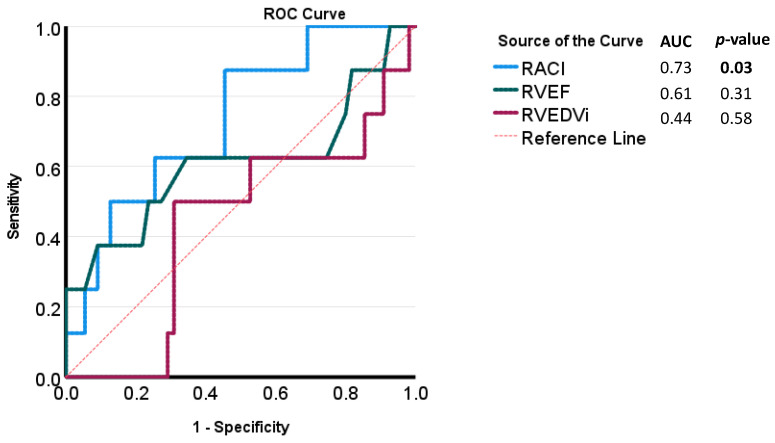
Receiver-operating characteristic (ROC) curves for CMR parameters for the evaluation of high-risk Importance Factor Scores (>2).

**Table 1 children-10-00400-t001:** Characteristics of the study population (N = 63) *.

Age at Complete TOF Repair (years)	0.5 (0.4, 1.0)
Age at CMR (years)	16.4 (12.9, 18.7)
Sex	
Male	37 (59%)
Female	26 (41%)
TOF Diagnosis	
Pulmonary Atresia	16 (25%)
Pulmonary Stenosis	47 (75%)
History of BTT Shunt	
Yes	9 (14%)
No	54 (86%)
History of Pulmonary Valve Replacement	
Yes	31 (49%)
No	32 (51%)
QRS Duration on EKG	134.0 (108.0, 149.0)
CMR Data ^†^	
LV End-Diastolic Volume (mL/m^2^)	76.2 (66.3, 85.9)
LV End-Systolic Volume (mL/m^2^)	31.2 (26.0, 36.5)
LV Ejection Fraction (%)	58.0 (54.0, 63.0)
RV End-Diastolic Volume (mL/m^2^)	119.1 (95.5, 138.3)
RV End-Systolic Volume (mL/m^2^)	61.3 (49.0, 73.2)
RV Ejection Fraction (%)	49.0 (44.0, 53.0)
Pulmonary Regurgitation Fraction (%)	39.0 (19.0, 51.5)
Minimum LA Volume (mL/m^2^)	10.4 (8.0, 13.3)
Maximum LA Volume (mL/m^2^)	24.4 (20.0, 30.2)
LA Ejection Fraction (%)	59.0 (49.1, 64.3)
Minimum RA Volume (mL/m^2^)	18.5 (12.6, 23.4)
Maximum RA Volume (mL/m^2^)	27.8 (22.0, 37.4)
RA Ejection Fraction (%)	34.9 (30.7, 45.8)
RA Coupling Index (%)	15.0 (10.5, 22.1)

* Data are represented as medians (25th %ile, 75th %ile) or n (%).^†^ N = 63 except LV Ejection Fraction (N = 61), Minimum LA Volume (N = 62), Minimum RA Volume (N = 62), and RA Coupling Index (N = 62). TOF = Tetralogy of Fallot, CMR = Cardiac Magnetic Resonance Imaging, BTT = Blalock–Thomas–Taussig, EKG = Electrocardiogram, LV = Left Ventricle, RV = Right Ventricle, LA = Left Atrium, RA = Right Atrium.

**Table 2 children-10-00400-t002:** Associations between atrial CMR measurements, additional CMR parameters, and age *^,†^.

	Minimum LA Volume (mL/m^2^)	Maximum LA Volume (mL/m^2^)	LA Ejection Fraction (%)	Minimum RA Volume (mL/m^2^)	Maximum RA Volume (mL/m^2^)	RA Ejection Fraction (%)	RA Coupling Index (%)
CMR Parameters							
LV End-Diastolic Volume (mL/m^2^)	0.26, 0.04 ^‡^	0.50, <0.0001 ^‡^	0.18, NS	0.31, 0.02 ^‡^	0.40, 0.001 ^‡^	0.09, NS	0.17, NS
LV End-Systolic Volume (mL/m^2^)	0.29, 0.02 ^‡^	0.38, 0.003 ^‡^	0.01, NS	0.35, 0.005 ^‡^	0.28, 0.03 ^‡^	−0.24, NS	0.24, NS
LV Ejection Fraction (%)	−0.18, NS	0.03, NS	0.29, 0.02 ^‡^	−0.12, NS	0.03, NS	0.28, 0.03 ^c^	0.04, NS
RV End-Diastolic Volume (mL/m^2^)	0.00, NS	0.08, NS	0.05, NS	−0.06, NS	−0.04, NS	0.06, NS	
RV End-Systolic Volume (mL/m^2^)	0.07, NS	0.05, NS	−0.12, NS	0.03, NS	−0.01, NS	−0.02, NS	−0.36, 0.004 ^‡^
RV Ejection Fraction (%)	−0.07, NS	0.13, NS	0.30, 0.02^‡^	−0.09, NS	0.06, NS	0.15, NS	−0.01, NS
Pulmonary Regurgitation Fraction (%)	−0.13, NS	−0.16, NS	−0.03, NS	−0.21, NS	−0.24, NS	0.03, NS	−0.45, 0.0002 ^‡^
Minimum LA Volume (mL/m^2^)		0.62, <0.0001 ^‡^	−0.61, <0.0001 ^‡^			−0.30, 0.02 ^‡^	0.30, 0.02 ^‡^
Maximum LA Volume (mL/m^2^)			0.12, NS			−0.01, NS	0.37, 0.003 ^‡^
LA Ejection Fraction (%)							−0.03, NS
Minimum RA Volume (mL/m^2^)	0.37, 0.003 ^‡^	0.50, <0.0001 ^‡^	−0.01, NS		0.78, <0.0001 ^‡^	−0.46, 0.0002 ^‡^	
Maximum RA Volume (mL/m^2^)	0.22, NS	0.57, <0.0001 ^‡^	0.24, NS			0.03, NS	0.63, <0.0001 ^‡^
RA Ejection Fraction (%)			0.38, 0.002 ^‡^				−0.42, 0.0007 ^‡^
Age at Complete TOF repair	0.23, NS	0.25, NS	−0.03, NS	0.17, NS	0.26, 0.04 ^‡^	−0.04, NS	0.27, 0.03 ^‡^
Age at CMR	0.34, 0.006 ^‡^	0.32, 0.01 ^‡^	−0.11, NS	0.36, 0.005 ^‡^	0.38, 0.002 ^‡^	0.01, NS	0.40, 0.001 ^‡^

* Results are expressed as Spearman ρ coefficients, *p*-values. *^†^* N = 63 for all CMR parameters except LV Ejection Fraction (N = 61) and Minimum RA Volume (N = 62). ^‡^ *p*-value < 0.05 is considered statistically significant. CMR = Cardiac Magnetic Resonance Imaging, LV = Left Ventricle, RV = Right Ventricle, LA = Left Atrium, RA = Right Atrium, TOF = Tetralogy of Fallot, NS = Non-Significant.

**Table 3 children-10-00400-t003:** Associations between atrial CMR measurements and patient characteristics.

	N	LA Ejection Fraction (%)		Minimum RA Volume (mL/m^2^)		Maximum RA Volume (mL/m^2^)		RA Ejection Fraction (%)		RA Coupling Index (%)	
		Median (25th %ile, 75th %ile)	*p-*Value *	Median (25th %ile, 75th %ile)	*p-*Value *	Median (25th %ile, 75th %ile)	*p-*Value *	Median (25th %ile, 75th %ile)	*p-*Value *	Median (25th %ile, 75th %ile)	*p-*Value *
TOF Diagnosis			NS		0.03 *		NS		NS		0.02 *
Pulmonary Atresia	16	56.8 (45.7, 66.5)		20.6 (16.3, 29.1)		31.0 (27.7, 40.9)		32.2 (25.2, 40.7)		22.8 (15.1, 27.0)	
Pulmonary Stenosis	47	59.0 (49.1, 63.5)		14.76 (12.0, 23.2)		26.8 (20.4, 42.2)		35.5 (31.6, 47.1)		13.2 (9.8, 18.8)	
History of BTT Shunt			NS		NS		NS		NS		NS
Yes	9	59.7 (50.1, 62.4)		24.8 (13.0, 32.4)		37.1 (25.3, 50.6)		38.6 (26.4, 41.2)		19.5 (8.9, 28.6)	
No	54	58.3 (47.7, 65.8)		16.6 (12.4, 22.7)		27.6 (20.5, 35.7)		34.3 (30.8, 46.1)		14.9 (10.6, 22.1)	
History of PVR											
Yes: Pre-CMR	6	56.3 (50.3, 67.1)	NS^†^	32.8 (12.2, 41.0)	NS ^†^	54.9 (29.6, 61.7)	0.03 *^,†^	44.7 (34.2, 53.2)	NS^†^	33.2 (12.9, 48.9)	NS ^†^
Yes: Post-CMR	25	57.3 (46.6, 62.4)	0.03 *^,‡^	16.7 (10.8, 20.3)	NS ^‡^	27.5 (19.8, 35.1)	NS ^‡^	35.2 (30.5, 44.4)	NS^‡^	12.1 (7.7, 16.9)	NS ^‡^
No	32	60.4 (47.9, 71.6)		17.7 (12.8, 23.3)		27.7 (20.9, 37.2)		33.8 (29.5, 44.6)		17.0 (10.9, 22.8)	

* *p*-value < 0.05 is considered statistically significant. † *p*-value reflects comparison of pre-CMR PVR patients with the remainder of the cohort. ^‡^ *p*-value reflects comparison of post-CMR PVR patients with those who never underwent PVR (excludes the six patients who underwent PVR pre-CMR). CMR = Cardiac Magnetic Resonance Imaging, TOF = Tetralogy of Fallot, BTT = Blalock–Thomas–Taussig, PVR = Pulmonary Valve Replacement, LA = Left Atrium, RA = Right Atrium, NS = Non-Significant.

**Table 4 children-10-00400-t004:** Associations of atrial measurements with Importance Factor Scores and RV-Independent Scores *^,†^.

	Importance Factor Score			RV-Independent Score		
	≤2	>2	*p-*Value ^‡^	≤2	>2	*p-*Value ^‡^
CMR Parameters						
Minimum LA Volume (mL/m^2^)	10.3 (7.8, 13.1)	11.2 (7.9, 14.0)	NS	10.4 (7.8, 13.1)	10.3 (7.9, 14.4)	NS
Maximum LA Volume (mL/m^2^)	24.8 (19.2, 30.3)	24.4 (20.1, 30.3)	NS	23.8 (19.2, 30.0)	25.6 (20.1, 30.7)	NS
LA Ejection Fraction (%)	59.5 (51.0, 65.7)	53.6 (44.9, 62.4)	NS	59.4 (51.0, 65.1)	53.6 (46.3, 64.7)	NS
Minimum RA Volume (mL/m^2^)	16.6 (11.3, 23.2)	19.8 (14.3, 28.9)	0.04 ^‡^	15.4 (11.3, 22.7)	20.6 (14.7, 28.9)	0.02 ^‡^
Maximum RA Volume (mL/m^2^)	27.2 (19.9, 36.5)	29.0 (22.8, 40.8)	NS	26.8 (19.9, 35.7)	30.8 (24.7, 41.6)	NS
RA Ejection Fraction (%)	35.2 (31.2, 47.3)	33.3 (28.7, 40.0)	NS	35.3 (31.2, 47.3)	33.3 (28.7, 41.2)	NS
RA Coupling Index (%)	13.5 (9.9, 19.6)	18.6 (11.8, 26.8)	0.03 ^‡^	13.1 (9.8, 18.9)	19.5 (13.4, 26.8)	0.02 ^‡^

* Results are expressed as medians (25th %ile, 75th %ile). ^†^ N = 42 for those with score ≤ 2 and N = 21 for those with score >2, with the exception of minimum RA volume, where N = 41 for those with score ≤ 2 and N = 21 for score >2. ^‡^ *p*-value < 0.05 is considered statistically significant. RV = Right Ventricle, CMR = Cardiac Magnetic Resonance Imaging, LA = Left Atrium, RA = Right Atrium, NS = Non-Significant.

## Data Availability

The data presented in this study are available on request from the corresponding author. The data are not publicly available due to privacy restrictions.

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
