# Peer review of "The Use of Automated Atrial CMR Measures and a Novel Atrioventricular Coupling Index for Predicting Risk in Repaired Tetralogy of Fallot"

_children, 2023, doi:10.3390/children10020400_

Round 1

Reviewer 1 Report

The article is well-written and structured in the usual manner. All references are relevant.

The main advantage is a proposal of the new parameter (RACI - right atrioventricular index), which could be used for the assessment of right heart function in repaired TOF patients.

From my point of view, the presentation of results is a bit confusing with too many parameters. However, all covered parameters are relevant to the research. After deeper studying of the tables, the content is more understandable and could be presented in this form. 

Author Response

Thank you for your comments. I attempted to make some edits to the results section to make it more clear and concise. I deleted some of the less vital results from the text to avoid overwhelming the reader.

Reviewer 2 Report

The aim of the study was to examine the relationship between left and right atrial volumes and function with ventricular parameters on CMR, to determine the association of atrial measurements and atrioventricular coupling with the risk of adverse long-term outcomes by using an established risk score and to assess the association of atrial CMR parameters with pulmonary valve replacement.

The article is very interesting and well written. I suggest small improvements, in particular:

- improve the drafting of the abstract by organizing it into purpose, materials and methods, results and conclusion;

- in the materials and methods I suggest to remove the verb “we sought”;

- the results are very interesting but I suggest to  improve its writing.

Author Response

1. I edited the abstract to include the main objective of the study and organized it as you suggested
2. I removed the verb “we sought” from the sentence you referenced
3. I edited the results to make it more clear and concise. I tried to delete extraneous information to focus the reader’s attention on the more important findings.

Reviewer 3 Report

A very interesting article.

the article is very interesting. there are no major corrections. the only note concerns paragraph 3.5. you could specify the time elapsed between PVR (six patients ) and CMR. in the case of these patients in fact the time elapsed I assume is short given the results obtained.

in paragraph 4 (discussion) line 254-256 in this regard it would be interesting to add a comment to the results obtained.

Author Response

To answer your question regarding paragraph 3.5, for those 6 patients that underwent PVR prior to their CMR, the average duration from PVR to time of CMR was 6.6 years. For the 25 patients who underwent PVR after CMR, the average duration from CMR to PVR was 1.1 years. I have added a sentence in the results (paragraph 3.5) describing the elapsed time for the 6 patients as you suggested.